# The Association Between Serum Drug Concentration and a Flare in Rheumatoid Arthritis Patients Tapering TNF Inhibitors

**DOI:** 10.3390/ph18101506

**Published:** 2025-10-08

**Authors:** Zohra Layegh, Femke Hooijberg, Laura Boekel, Agnes E. M. Looijen, Elise van Mulligen, Floris C. Loeff, Lisanne Dijk, Radboud J. E. M. Dolhain, Theo Rispens, Gertjan J. Wolbink, Pascal H. P. de Jong

**Affiliations:** 1Amsterdam Rheumatology and Immunology Center, Location Reade, Department of Rheumatology, Dr. Jan van Breemenstraat 2, 1056 AB Amsterdam, The Netherlands; f.hooijberg@reade.nl (F.H.); l.boekel@reade.nl (L.B.); g.wolbink@reade.nl (G.J.W.); 2Department of Rheumatology, Erasmus MC, Dr. Molewaterplein 40, 3015 GD Rotterdam, The Netherlandselise.vanmulligen@erasmusmc.nl (E.v.M.); r.dolhain@erasmusmc.nl (R.J.E.M.D.); p.h.p.dejong@erasmusmc.nl (P.H.P.d.J.); 3Department of Rheumatology, Leiden University Medical Center, Albinusdreef 2, 2333 ZB Leiden, The Netherlands; 4Sanquin Diagnostic Service, 1066 CX Amsterdam, The Netherlands; f.loeff@sanquin.nl (F.C.L.); l.dijk@sanquin.nl (L.D.); t.rispens@sanquin.nl (T.R.); 5Sanquin Research and Landsteiner Laboratory Academic Medical Center, Department of Immunopathology, Plesmanlaan 125, 1066 CX Amsterdam, The Netherlands

**Keywords:** serum drug levels, tapering TNF inhibitors, anti-drug antibodies

## Abstract

**Objectives:** To assess the association between serum concentrations of adalimumab (ADL) and etanercept (ETN) and the occurrence of a flare in rheumatoid arthritis (RA) patients who are tapering methotrexate (MTX) or their TNF inhibitor. In addition, we explored the impact of tapering MTX on immunogenicity in patients with longstanding ADL use. **Methods:** ADL and ETN serum concentrations and anti-drug antibodies (ADAs) quantified with a drug-tolerant assay were determined in all RA patients who participated in the TARA trial. Within the TARA trial, two tapering strategies were compared, namely gradually tapering MTX followed by tapering a TNF-inhibitor (ADL or ETN) or vice versa. **Results**: In the current analysis, 111 RA patients who strictly followed the tapering strategy and had >3 blood samples were included, of them 41% tapered ADL and 59% tapered ETN. Both ADL and ETN concentrations decreased during tapering and stopping, but ADL was longer detectable after cessation compared to ETN. If MTX was tapered first, more ADAs against ADL were detectable in the serum, but it did not affect the serum concentrations. **Conclusions:** Our data showed that the majority of flares occur when the median serum concentration of ADL and ETN falls below 1 mg/L. If MTX is tapered first, there is a notable increase in the detection of ADAs, but this does not impact the median ADL serum concentration.

## 1. Introduction

Due to the advancement in management strategies and the increased utilization of biologic disease-modifying antirheumatic drugs (bDMARDs), approximately 60% of rheumatoid arthritis (RA) patients can now achieve sustained remission [1]. Current recommendations recommend tapering of DMARDs in this subgroup of RA patients, not only due to the prolonged absence of inflammation, but also to mitigate the risk of adverse events and reduce healthcare expenditures.

However, previous tapering trials for tumor necrosis factor inhibitors (TNFis) in RA have shown that 51–77% of patients experience a disease flare during tapering and discontinuation of the drug [2,3,4,5]. The occurrence of a disease flare has a significant long-term impact on patients’ lives [6]. On the other hand, 80% of patients are able to reduce their TNFi dosage and ~20% of patients can taper their TNFi to complete cessation while maintaining remission or low disease activity [7]. Although this makes tapering feasible, the occurrence of disease flares is still cumbersome. In order to prevent disease flares, it is important to know below which drug concentration a flare might be expected.

Recently, the EULAR points to consider for therapeutic drug monitoring (TDM) were published, in which they recommended TDM in specific clinical situations, such as an inadequate treatment response or, conversely, when patients are in a state of low disease activity or remission [8]. Despite the observed correlation between serum concentrations and clinical response, there are no well-designed studies that have investigated the therapeutic range or the minimum effective concentration. Therefore, the recommendation discourages reliance on these concentrations for dosing purposes. Most of the available evidence on the therapeutic range is based on studies in RA patients who were treated with adalimumab (ADL). In patients treated with ADL, a general trend emerges where those with a more favorable treatment response or maintaining lower disease activity tend to exhibit higher drug concentrations in their blood, although the curve plateaus at trough levels around 5 mg/L [9]. An open-label non-inferiority study by l’Ami et al. found that RA patients treated with ADL, having trough concentrations >8 mg/L, could extend their standard dosing interval to once every 3 weeks without experiencing a disease flare [10]. However, no previous study has linked drug levels to flares during tapering. While TDM is only recommended in selected clinical scenarios, the minimum effective concentration during tapering, a key parameter for successful de-escalation, remains poorly defined. Therefore, this study aims to address that knowledge gap.

Furthermore, limited information is available regarding the influence of methotrexate (MTX) on anti-drug antibodies (ADA) in patients using ADL in conjunction with MTX over an extended duration. The extended survival of TNFi observed with co-administration of MTX is attributed to a likely reduction in ADA formation [11,12]. This observation aligns with Krieckaert et al. and others, who demonstrated that RA patients concurrently using ADL and MTX had elevated ADL concentrations and experienced less ADA formation, particularly notable during the initial stages of treatment [13,14]. Nevertheless, the impact of MTX in the later stages of treatment on the detection of ADAs against TNFi remains uncertain.

In summary, our knowledge of TNFi pharmacology during tapering, including ADA formation, remains incomplete. Therefore, the aim of this study is to examine the link between ADL and etanercept (ETN) serum concentrations and the occurrence of flares in well-controlled RA patients who are gradually tapering their DMARDs to cessation, using data from the TARA trial [7]. The original TARA trial was set up to evaluate the efficacy of two different tapering strategies, namely gradual tapering of conventional synthetic DMARDs followed by TNFi and vice versa. In addition, we explored the impact of tapering MTX on immunogenicity in patients with longstanding use of ADL. We hypothesized that the majority of flares would occur as serum drug concentrations fall below a specific threshold, and that immunogenicity would be influenced by the order of tapering.

## 2. Results

### 2.1. Patients

In the original TARA trial, 189 RA patients with a well-controlled disease were included. For this analysis, we selected all RA patients who strictly followed the tapering protocol and who had >3 samples available (*n* = 111). Demographic and clinical characteristics were comparable between included and excluded patients (see Appendix A). Of the 111 included patients, 22 tapered adalimumab (ADL) first, followed by methotrexate (MTX), 24 tapered MTX first followed by ADL, 32 tapered the etanercept (ETN) first followed by MTX and 33 tapered MTX first, followed by ETN (see Appendix A). The demographic and clinical characteristics are shown in Table 1. The included RA patients had a median disease duration of 5.7 years and were mostly female (69%) with a mean (SD) age of 57 (11.3) years. At baseline, the mean (SD) DAS was 1.0 (0.49) and median (IQR) MTX dose was 20 (15–25) mg/week.

### 2.2. ADL and ETN Serum Concentrations During Tapering

In RA patients who tapered TNFi first, the median (IQR) ADL and ETN serum concentrations before tapering initiation were 9.5 (5.5–10.5) and 2.3 (1.8–3.0) mg/L, respectively. For those who tapered MTX first, the concentrations were 7.4 (4.7–9.8) and 2.4 (1.9–3.2) mg/L. During gradual tapering of ADL and ETN, the drug concentration also declined (Figure 1A and Figure 2A). The median (IQR) for complete ADL elimination from the circulation after cessation was 12 (9.3–14.7) months, while it took 3.0 (1.3–4.7) months for ETN to be undetectable in the serum (*p* = 0.001), (see also Appendix A). If a flare occurred, the TNFi dosage was intensified, which also resulted in a noticeable increase in serum concentration (Figure 1B and Figure 2B). Scatter plot of serum concentration of ADL and ETN in patients with and without a flare, in which a flare is set at T0, are given in Appendix A. The individual serum concentrations of ADL and ETN in patients who experienced a flare and restarted treatment, according to the course of the trial, are given in Appendix A.

### 2.3. ADL and ETN Serum Concentrations During a Flare

At the moment of a flare, the median (IQR) serum concentration for ADL was 0.44 (0.11–1.15) mg/L and for ETN was 0.05 (0.05–1.29) mg/L. The ADL serum concentration at the moment of flare was significantly lower than the concentration 3 months prior to a flare with a median (IQR) of (0.44 (0.11–1.15) vs. 2.03 (0.98–3.53) mg/L (*p* = 0.036, Figure 1B,D). Although we found a trend toward significance in the ETN serum concentrations at the moment of a flare compared to the serum concentrations 3 months prior to a flare, this difference was very small with a median (IQR) of 0.05 (0.05–1.29) vs. 0.94 (0.22–2.99) mg/L (*p* = 0.05, Figure 2B,D). These GEE analyses showed that higher ADL levels were associated with a reduced risk of a flare (odd ratios 0.29 for high versus low ADL levels, 95% CI:0.09–0.92, *p* = 0.036 and 0.37 for high versus medium,95% CI: 0.14 0.98, *p* = 0.046) (Table 2). This relationship was not observed for etanercept (Table 2).

### 2.4. ADAs Targeting ADL

There appears to be a higher detection of anti-drug antibodies (ADAs) in patients tapering MTX first compared with those tapering ADL first (Figure 3A). However, the difference is not statistically significant after 24 months (*p* = 0.145). Although ADA detection increased if MTX was tapered first, this did not seem to affect ADL serum concentrations (Figure 3B).

## 3. Discussion

### 3.1. Main Findings

In this study, we investigated the serum concentrations of TNF inhibitors (TNFis) in rheumatoid arthritis (RA) patients with a well-controlled disease who gradually tapered their medication to complete cessation. Our data showed that the majority of flares occur when the median serum concentration of ADL and ETN falls below 1 mg/L. At the moment of flare, the median concentration of ADL and ETN was 0.44 mg/L and 0.05, respectively. In addition, we observed a trend towards increased ADA positivity in patients who tapered their MTX first, which was not observed in patients who tapered their ADL first. However, the increase in ADAs did not affect the serum concentration of ADL.

Previous well-defined studies that attempted to determine a therapeutic range or a minimum effective concentration are scarce. Most of the evidence is centered around individuals with RA receiving ADL treatment, suggesting that a substantial clinical benefit becomes apparent in the majority of patients when their serum concentrations reach a minimum of 1 mg/L [15,16,17,18]. In our study, the median serum concentration during a flare seems lower than 1 mg/L for ADL as well as ETN. Most of the patients experienced a flare at a median concentration of 0.44 mg/L, with an upper interquartile range (IQR) limit of 1.15 mg/L for ADL and 1.29 for ETN. To prevent flares in any patient, it is advisable to maintain serum concentrations within the therapeutic range. With our data, we show that there is an association between serum concentration and a flare in patients tapering ADL. However, it is crucial to emphasize that the occurrence of a flare is not solely determined by target blockade. Our findings are most relevant to well-controlled RA patients on long-term combination therapy. Generalizability to other groups, such as patients with shorter disease duration, those on monotherapy, or individuals with other TNF inhibitor-treated diseases, should be interpreted with caution, given differences in treatment response, disease course, and underlying pathophysiology. This complexity is reflected by the wide confidence intervals, and prevents us from determining the exact minimal concentration at which a flare may be prevented.

### 3.2. Pharmacokinetics

Although ADL and ETN both bind to and neutralize TNF, they differ structurally and display distinct pharmacokinetic properties and binding characteristics to TNF [19,20]. ADL, for example, is a fully humanized monoclonal anti-TNFα antibody, while ETN is a soluble p75 TNF receptor Fc fusion. Both TNFis possess an Fc region that binds to the Fc receptor. The Fc region of TNFi plays a crucial role in their clearance from the body. Variations in the Fc regions among different TNFis, like ADL, contribute to differences in their half-lives. This discrepancy is partly due to their interactions with the neonatal Fc receptor (FcRn), which protects antibodies from degradation and thereby recycles ADL back into the circulation, extending its half-life. In contrast, ETN is recycled less efficiently through FcRn, contributing to its faster clearance. Unlike traditional monoclonal antibodies, etanercept takes a unique form as a soluble TNF receptor fusion protein. It combines the extracellular ligand-binding segment of the human TNF receptor (TNFR) with the Fc region of human immunoglobulin G1 (IgG1). This distinctive structure alters the role of the Fc region and its interaction with Fc receptors in ETN mechanism of action. These differences may contribute to the variation in half-life observed between ADL and ETN [21,22,23].

It is already established that ADL exhibits a longer half-life than ETN with values of 14 and 3 days, respectively [24,25]. However, data on the clearance of TNFis after cessation is scarce. Berkhout et al., for example, showed that even after 24 weeks, ADL remained in complex with TNF following either the prolongation or discontinuation of ADL treatment [26]. ETN, on the other hand, is eliminated much faster from the bloodstream. The difference in clearance rates may partly explain why lower drug levels were associated with an increased flare risk for ADL, but not ETN. In this study, tapering and blood sampling occurred every 3 months, regardless of the dosing interval. Due to ETN’s shorter half-life, patients can rapidly reach low drug levels, even among those who stay in remission without having a flare. As a result, many patients without flares also fall into the ‘low concentration’ category, which makes it statistically difficult to demonstrate a differential association between ETN levels and flare risk. In contrast, the longer half-life of ADL causes a slower decline in drug concentrations, leaving more patients around the threshold level for a longer period of time. This makes the association with flares more apparent, even with the suboptimal timing of blood sampling. Therefore, difference in clearance should be carefully considered when implementing tapering strategies in daily practice.

In addition to variations in pharmacokinetics, there are also disparities in immunogenicity.

### 3.3. Immunogenicity

Therapeutic antibodies, including fully human ones, can trigger undesired immune responses due to the presence of foreign determinants like the antigen-binding site, known as the idiotype. Prior research indicates that the immune response, particularly the development of ADAs, is typically confined to this idiotype [27,28,29]. Because ETN is the only TNFi without an idiotype, it is considered non-immunogenic [30,31]. Moreover, previous research suggests that patients who are treated with a higher TNFi dosage also have less ADA formation, which is known as “high dose tolerance” [32]. Lastly, combining TNFis with csDMARDs, especially MTX, also reduces the formation of ADAs which seems to be dose-dependent [33]. MTX can modulate the immune response to TNFi, which is particularly crucial in the initial stages of treatment, by suppressing the expansion of early T- and B-cell populations. T- and B-cells play a critical role in generating ADAs against TNFi. However, in patients undergoing long-term treatment, the immune system develops tolerance. Over time, many individuals receiving TNFi therapy experience immune tolerance, characterized by a decrease in drug immunogenicity and ADA responses. This phenomenon is attributed to mechanisms such as immune exhaustion, clonal deletion, or the induction of regulatory pathways [34,35].

In the earlier stages of treatment, the formation of ADAs against ADL is associated with lower ADL concentrations and a lower likelihood of achieving remission [13]. Whether the protective effect of MTX on ADA formation endures with prolonged use or in later stages of the treatment is unknown. To our knowledge, this is the first study that analyzed the effect of gradual tapering of MTX on ADA detection in well-controlled RA patients with prolonged ADL usage. We observed a trend towards increased ADA positivity in patients who tapered MTX first, which was not observed in patients who tapered ADL first. However, this did not appear to influence ADL serum concentrations. This finding could be explained by the utilization of a drug-tolerant assay in this study, which detects all types of ADAs, including those that are not clinically relevant. It is, therefore, plausible that the ADAs observed in this long-term cohort were predominantly non-neutralizing or of low affinity. This interpretation is supported by previous data showing that ADA formation generally stabilizes after the first six months of treatment, suggesting that later-detected ADAs are unlikely to be functionally significant, reinforcing the guideline to only test drug levels and ADAs on indication.

Taken together, these findings indicate that tapering MTX later in treatment has minimal clinical impact on ADL effectiveness, while MTX-related side effects could potentially be reduced [8,16,34,36].

### 3.4. Strengths and Limitations

We are the first to investigate the association between serum concentrations and the onset of a flare during the tapering process. Our results could contribute to the development of a more personalized tapering approach. Strengths of the study include the data collection and the use of a large multicenter, single-blinded, randomized controlled trial that was conducted across 12 rheumatology centers, representing daily practice. This is also the first tapering trial with a follow-up period of 24 months, in which the occurrence of a flare is well defined and registered.

A limitation of this study is that it is a post hoc analysis of the TARA trial with a limited number of patients per tapering strategy and thus the results should be interpreted carefully. To minimize the chance of false positive tests, we prespecified our outcomes and analyses. Moreover, the exact moment of the disease flare was difficult to measure, because of the study design that included 3-monthly visits. Furthermore, serum samples were collected at non-trough time points and no correction for exact interval between the last injection was applied. This introduces additional variability in drug concentrations and could, in theory, bias the comparison between patients with and without a flare. Although we cannot completely exclude this possibility, sampling occurred at fixed 3-monthly visits independent of disease activity, making systematic differences between groups unlikely. Importantly, the observed association is in line with our predefined hypothesis, which strengthens the validity of our findings despite this limitation. Also, our flare definition, namely DAS > 2.4 or SJC > 1, is based on expert opinion and in our opinion reflects the need to intensify treatment in daily practice [37]. The association between a flare and drug levels does not take the patient’s perspective into account and it is known that the disease burden already worsens 3 months prior to a flare, which might have implications for the therapeutic range [6]. Lastly, the exclusion of 78 patients due to <3 available samples, drop out, and protocol violations could potentially introduce a selection bias. However, no differences in baseline characteristics were observed between included and excluded patients, which suggests that the used study population is a representative of the total TARA trial.

## 4. Materials and Methods

### 4.1. Patients

For this study, we selected all patients who participated in the TApering strategies in Rheumatoid Arthritis (TARA) trial (NTR2754) who had ≥3 available samples at different 3-monthly study visits and strictly followed the tapering protocol. In addition, RA patients who were using >1 csDMARD were excluded from the analysis. (see Appendix A). The TARA trial, a multicenter, single-blinded, randomized controlled trial, was conducted across 12 rheumatology centers in the southwestern region of the Netherlands. Adult RA patients with a well-controlled disease, defined as a disease activity score (DAS) ≤2.4 and a swollen joint count (SJC) ≤1 [14], at 2 consecutive visits, which were at least 3 months apart, who were treated with both a csDMARD, methotrexate (MTX), and a TNFi, adalimumab (ADL), or etanercept (ETN), were included. Patients were randomized into gradual tapering their MTX in the first year, followed by gradual tapering of ADL or ETN in the second year or vice versa. The total tapering schedule of MTX as well as the TNFis took 6 months, with dose adjustments every 3 months as long as no disease flares occurred. The MTX dose was first halved, then quartered, and thereafter it was stopped. ADL and ETN were gradually tapered. Tapering was commenced by doubling the dosing interval, followed by halving the dose, and finally it was stopped (see Supplementary Text S1). If a disease flare occurred, defined as DAS > 2.4 or SJC > 1, tapering was stopped and patients reverted to their previously effective treatment. If necessary, treatment was intensified, according to a treat-to-target approach, until low disease activity was achieved. After a flare, no further attempts were made to taper medication for the remainder of the study. There were no constraints on the use of non-steroidal anti-inflammatory drugs or intra-articular glucocorticoid (GC) injections during follow-up. However, before inclusion, patients were asked not to use oral GCs. Medical ethics committees of each participating center approved the protocol and all patients provided written informed consent before inclusion. Further details of the study can be found elsewhere [7].

### 4.2. Outcome Measures and Assessments

RA patients were assessed at baseline and every 3 months thereafter (0, 3, 6, 9, 12, 15, 18, 21, and 24 months). At each study visit, patients were seen by the research nurse, who calculated the DAS, and additional blood samples were taken after which the sera were stored at −80 degrees. The blood samples were not collected on a specific day and were not taken at trough levels.

### 4.3. Measurement of Serum Drug Concentrations

The collected serum samples were thawed before the measurement of the ADL and ETN serum concentrations. The ADL and ETN serum concentrations were measured using a sandwich enzyme-linked immunosorbent assay (ELISA) developed by Sanquin Services Amsterdam, which is based on the principle that ADL and ETN are captured through their ability to bind TNF (see Appendix A). The detection level of the assay for ADL was 0.01 mg/L and for ETN, 0.1 mg/L. Further details on these assays can be found elsewhere [9,38,39]. Firstly, changes in ADL and ETN concentrations during tapering were explored in RA patients with and without a flare. Subsequently, the elimination of ADL and ETN (after cessation) was investigated in RA patients without a flare who tapered their TNFi first. Finally, the critical drug concentration of ADL and ETN at which a flare occurred was investigated through comparison of the TNFi serum concentration at the moment of flare with the serum concentration 3 months prior to a flare in RA patients who experienced a flare. All serum samples were non-trough and no statistical correction was applied for the interval between the last injection and the blood draw.

### 4.4. Measurement of Anti-Drug Antibodies (ADAs)

In RA patients who tapered ADL, ADAs were determined. ADAs were not measured in patients who tapered ETN, because it is well known that the development of ADAs does not significantly impact the clearance of ETN, since ETN has a very low immunogenicity [40]. The measurement of ADAs that target ADL was conducted with a drug-tolerant acid-dissociation lanthanide-fluorescence immunoassay (ALFIA) [41]. In ALFIA, ADA–drug complexes are dissociated, which is followed by their binding to a drug derivative labeled with europium. Subsequently, an IgG pulldown is performed on Sepharose beads. After elution of europium, detection is accomplished by measuring time-resolved fluorescence originating from europium chelate complexes (see Appendix A).

### 4.5. Statistics

Demographic and clinical characteristics are presented as mean (standard deviation, SD) or median (interquartile range, IQR) depending on whether the distribution is normal or not. We compared baseline characteristics between included and excluded patients to evaluate the potential impact of selection bias. A survival analysis was performed to examine how long it takes before ADL and ETN drug levels are unmeasurable in the blood circulation after their cessation. If ADL concentrations were below the detection level of the ADL assay (<0.01 mg/L), we choose a level of 0.01 mg/L. If ETN concentrations were undetectable (<0.1 mg/L), we choose a level of 0.05 mg/L.

Secondly, the minimum serum concentration to possibly prevent a disease flare was assessed by comparing the median serum concentration of ADL and ETN at the moment of a flare with the serum concentrations 3 months prior to a flare. To further explore the relationship between flare (outcome variable) and drug level, generalized estimated equations (GEE) with a logit link function and binomial distribution were used. An exchangeable correlation structure was applied to account for within-subject correlation. Separate models were constructed for ADL and ETN. Since no linear relationship was observed between drug level and flare in either model, drug level cut-offs were based on the drug level distribution tertiles. Cut-off values for ADL were <1.81 mg/L (low), 1.81–6.06 mg/L (medium), and >6.06 mg/L (high). For ETN, the cut-off values were <0.283 mg/L (low), 0.283–2.24 mg/L (medium), and >2.24 mg/L (high). Finally, the association between ADA formation and tapering strategy was explored. We compared the proportion of RA patients with detectable ADAs after 24 months of follow-up between those who tapered the MTX first and those who tapered the ADL first. For this analysis we included patients till they experienced a flare, and we used a Fisher’s exact test. Descriptive statistics were used to analyze our data. Analyses were performed in Stata 17.0. R version 4.0.3 was used for the GEE analyses. *p*-values < 0.05 were considered statistically significant.

## 5. Conclusions and Future Directions

This study sheds light on the serum concentration during a flare and immunogenicity of TNF inhibitors in rheumatoid arthritis patients with a well-controlled disease who are tapering and stopping their DMARDs and provides insights for measuring drug concentration in daily practice. Therapeutic drug monitoring can help identify optimal dosing regimens that balance efficacy and safety, particularly in patients at risk of disease relapse or treatment failure during tapering. Our findings could serve as a basis for future research, e.g., prospective studies that test TDM-guided tapering strategies versus standard care, or investigations into the characteristics of ADAs detected by drug-tolerant assays in long-term treatment, with the ultimate goal of determining the TNFi concentration below which a flare can be expected, which, in our study, seems to be on average below 1 mg/L.

## Figures and Tables

**Figure 1 pharmaceuticals-18-01506-f001:**
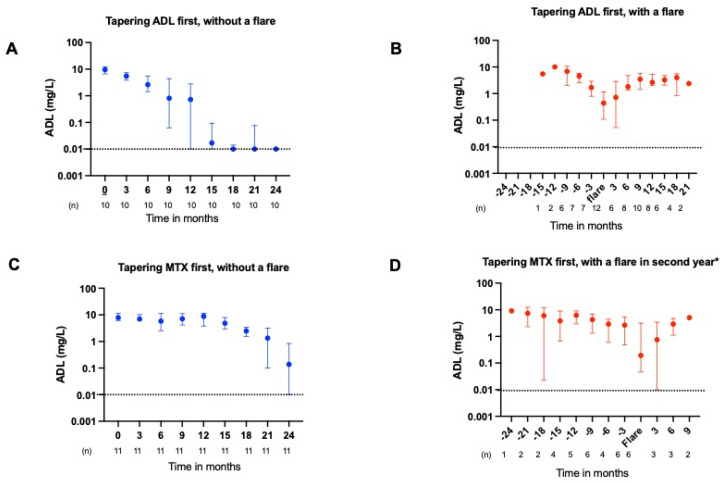
ADL drug levels in RA patients with and without a disease flare in different tapering strategies. (**A**) (patients without a flare) and (**B**) (patients with a flare) show the median ADL drug concentrations with an interquartile range of RA patients who tapered ADL in the first year and MTX in the second year. (**C**) (without a flare) and (**D**) (with a flare) show the median ADL drug concentrations with interquartile range, for RA patients who tapered MTX in the first year and ADL in the second year. The dotted line shows the detection limit of the ADL assay of 0.01 mg/L. Y-axis is logarithmic. * A total of 12 patients experienced a flare during tapering of MTX in the first year, while an additional 6 patients experienced a flare during tapering of ADL in the second year. *Abbreviations: ADL, adalimumab; MTX, methotrexate; RA, rheumatoid arthritis.*

**Figure 2 pharmaceuticals-18-01506-f002:**
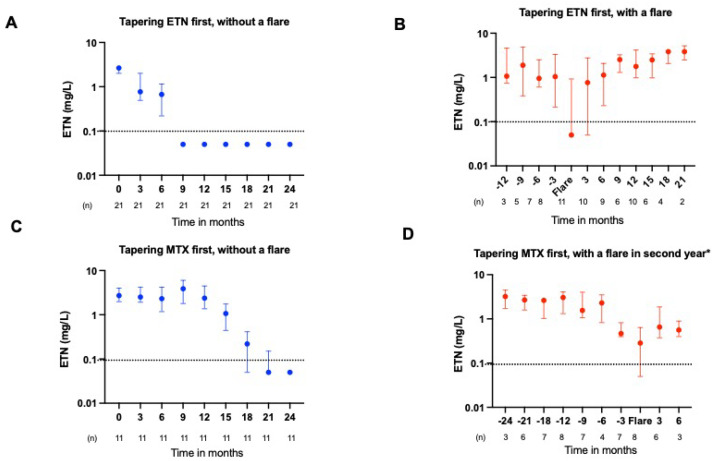
ETN drug levels in RA patients with and without a disease flare in different tapering strategies. (**A**) (patients without a flare) and (**B**) (patients with a flare) show the median ETN drug concentrations with an interquartile range of RA patients who tapered ETN in the first year and MTX in the second year. (**C**) (patients without a flare) and (**D**) (patients with a flare) show the median ETN drug concentrations with interquartile range, for RA patients who tapered MTX in the first year and ETN in the second year. The dotted line shows the detection limit of the ETN assay of 0.1 mg/L. Y-axis is logarithmic. * A total of 22 patients experienced a flare during tapering of MTX in the first year, while an additional 8 patients experienced a flare during tapering of ETN in the second year. *Abbreviations: ETN, etanercept; MTX, methotrexate; RA, rheumatoid arthritis.*

**Figure 3 pharmaceuticals-18-01506-f003:**
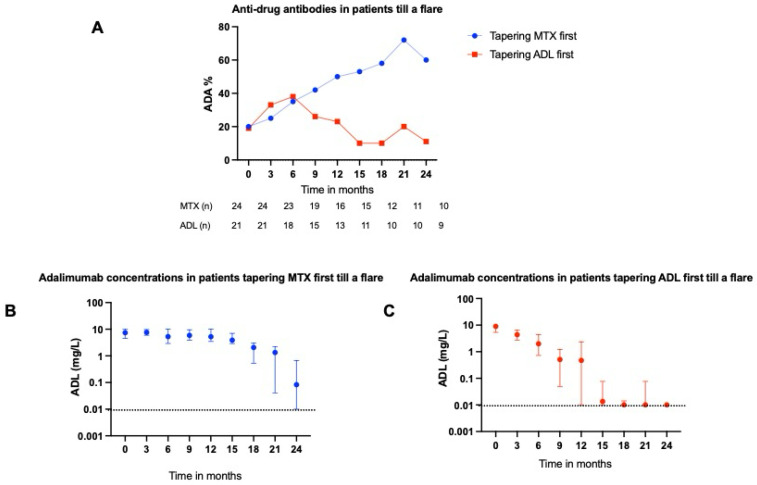
Anti-drug antibody detection in RA patients with different tapering strategies. (**A**) shows the proportion of RA patients with detectable percentages of ADAs over time. (**B**,**C**) show the median ADL serum concentrations with interquartile range of RA patients who tapered MTX in the first year and ADL in the second year until a disease flare occurred and vice versa, respectively. The dotted line shows the detection limit of the ADL assay of 0.01 mg/L. *Abbreviations: ADA, anti-drug antibody; ADL, adalimumab; MTX, methotrexate; RA, rheumatoid arthritis.*

**Table 1 pharmaceuticals-18-01506-t001:** Demographic and clinical characteristics of RA patients in the current analysis with different tapering strategies included in this study.

*Characteristics*	*All RA Patients* (*N = 111*)	*Tapering ADL First, Followed by MTX* (*N = 22*)	*Tapering MTX First, Followed by ADL* (*N = 24*)	*Tapering ETN First, Followed by MTX* (*N = 32*)	*Tapering MTX First, Followed by ETN* (*N = 33*)
*Demographic*					
*Age (years), mean (SD*)	57 (11.3)	59 (8.9)	52 (10.8)	58 (10.3)	58 (13.5)
*BMI, median (IQR*)	26 (24–29)	25 (25–30)	26 (24–28)	27 (24–30)	25 (23–29)
*Sex, female, n (%*)	69 (62)	11 (50)	17 (70)	20 (63)	21 (63)
Disease *duration (years), median (IQR*)	5.7 (3.9–8.5)	7.1 (4.2–9.0)	6.0 (3.1–9.1)	5.7 (4.0–8.5)	4.7 (2.5–70
*Clinical*					
*RF positive, n (%*)	56 (50)	13 (72)	12 (57)	16 (55)	15 (46)
*ACPA positive, n (%*)	70 (63)	12 (67)	15 (71)	20 (69)	23 (74)
*DAS44, mean (SD*)	1.0 (0.49)	1.0 (0.44)	1.1 (0.6)	1.0 (0.53)	0.9 (0.35)
*ESR, median (IQR*)	8 (3–15)	6 (2–18)	5.5 (2–12)	9.5 (4.5–16)	8 (5–14)
*CRP, median (IQR*)	2 (1–5)	2 (1–5)	1.8 (1–5)	2 (1–8)	3 (1–5)
*Experienced a flare, n (%*)	47 (42)	12 (54)	13 (54)	11 (34)	11 (33)
*Median MTX dose in mg/week, median (IQR*)	20 (15–25)	15 (14–25)	15 (10–25)	22 (15–25)	25 (15–25)
*Median TNFi serum concentration in mg/L, median (IQR*)	-	9.5 (5.5–10.5)	7.4 (4.7–9.8)	2.3 (1.8–3.0)	2.4 (1.9–3.2)

Abbreviations: ADL, adalimumab; BMI, body mass index; CRP, C-reactive protein; DAS, disease activity score; ESR, erythrocyte sedimentation rate; ETN, etanercept; IQR, interquartile range; MTX, methotrexate; SDs, standard deviations; TNFi, tumor necrosis factor inhibitor.

**Table 2 pharmaceuticals-18-01506-t002:** Relationship between drug levels of adalimumab and etanercept and the risk of a disease flare.

Adalimumab				Etanercept			
Drug Level (mg/L)	OR	(95% CI)	*p* Value	Drug Level (mg/L)	OR	(95% CI)	*p* Value
**Low *** (*<1.81*)	-	-	-	**Low *** (*<0.283*)	-	-	-
**Medium** (*1.81–6.06*)	0.29	(0.09–0.92)	** *0.036* **	**Medium** (*0.283–2.24*)	1.35	(0.47–3.92)	0.57
**High** (*>6.06*)	0.37	(0.14–0.98)	** *0.046* **	**High** (*>2.24*)	2.14	(0.86–5.32)	0.10

Table showing results of generalized estimated equations (GEEs) analyses. Data are presented as odds ratio (OR) with corresponding 95% confidence intervals (CIs) and *p*-values. The OR represents the odds of a flare occurring in each group relative to the reference group (*). *p*-values below 0.05 were considered statistically significant and are highlighted in bold.

## Data Availability

The data presented in this study are available on request from the corresponding author due to privacy concerns, as the dataset contains sensitive information from patients treated in a clinical setting. To protect patient confidentiality and comply with ethical and legal regulations, access is limited.

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
