# Peer review of "The Association Between Serum Drug Concentration and a Flare in Rheumatoid Arthritis Patients Tapering TNF Inhibitors"

_pharmaceuticals, 2025, doi:10.3390/ph18101506_

Round 1
Reviewer 1 Report
Comments and Suggestions for Authors
The manuscript by Layegh et al. presents a post hoc analysis of the TARA trial examining the association between adalimumab (ADA) and etanercept (ETA) serum concentrations during gradual tapering and disease flare risk, as well as the impact of methotrexate (MTX) tapering on anti-drug antibody (ADA) formation. The study found that lower TNFi trough levels, particularly ADL concentrations below ~1 mg/L, were linked to a higher risk of flares. It also found that MTX tapering may mildly increase ADA detection without significantly altering serum ADL levels.
The manuscript is well-written and uses an appropriate methodology, but the clinical significance claimed by the authors is perhaps overstated.
Major Points:
The analyses included only 111 patients with ≥3 serum samples who followed the tapering protocol. The impact of excluding 78 participants should be clarified; could their disease activity, adherence or baseline drug levels have differed? At least this point should be included as a limitation in the discussion section.
While the trend towards more ADA detection after MTX tapering is noted, the lack of statistical significance (p = 0.145) should be reflected in more conservative language.
Assuming Figure 1 has not been misinterpreted, 21 patients with ADA serum concentrations below 1 mg/L appear to have remained free of flares at 24 months, regardless of whether MTX was tapered before or after ADA. Conversely, 18 patients with similarly low ADA levels experienced a flare within the same timeframe. This corresponds to an approximate flare rate of 50% following ADA cessation. Considering this, the predictive value of circulating ADA concentrations for flare risk appears limited. Therefore, I would argue that the final sentence of the conclusion of this manuscript may overestimate the clinical utility of monitoring ADA levels in the context of RA drug tapering strategies.
Reviewer 2 Report
Comments and Suggestions for Authors
There is no novelty and scientific soundness in the research article and moreover the material and data are insufficient for publication, therefore i not recommended this article for publication.
Reviewer 3 Report
Comments and Suggestions for Authors
Add updated literature on serum drug concentration and steady drug concentration.
Add drug literature and also draw their structures.
In methodology section please mention the protocol that on which day you monitored the serum drug concentration.
What unit for serum drug concentration you used please mention.
Why you selected serum drug concentration instead of plasma drug concentration.
How you studied the tapering of methotrexate on immunogenicity.
You used abbreviation like AdA and Adl in abstract section please add full names
Reviewer 4 Report
Comments and Suggestions for Authors
This is a well-conducted and clearly reported study. The biggest improvements would come from tightening the abstract and discussion, clarifying methods (sampling & cut-offs), moderating claims about borderline/non-significant results and minor language polishing. A careful language edit would improve fluency. Acronyms (e.g., ADA, ADL, ETN, MTX) are used heavily—consider adding a table of abbreviations early or use fewer abbreviations in the main text.
Email address (preferably institutional): please remove.
- Abstract
- Please consider tightening the abstract. It currently packs too much numerical detail, which may overwhelm readers. Abstracts work best when numbers are limited to the most critical findings (e.g., serum threshold <1 mg/L, ADL detectable longer than ETN, ADA increase with MTX tapering first).
- Keywords are missing.
- Introduction
- Some sentences are dense and repetitive. For instance, the discussion on therapeutic ranges could be more concise.
- The transition from background to rationale is somewhat abrupt. Explicitly state the gap: "No previous study has linked drug levels to flares during tapering."
- A schematic summarizing “known to….unknown… to study aim” could make the rationale sharper.
- Methods
- Clarify whether serum samples were taken at trough levels or not. It’s mentioned they were “non-trough,” but the implication for variability in results could be explained earlier.
- Statistics: The cut-offs (e.g., ADL <1.81, 1.81–6.06, >6.06 mg/L) are reasonable, but how were they chosen? Arbitrary tertiles, or based on prior studies? State rationale.
- Results
- Some numbers are repeated unnecessarily (e.g., flare concentrations cited multiple times in both text and figures). Consider trimming redundancy.
- The borderline significance for ETN (p=0.05) is over-emphasized—reframe cautiously (“trend toward significance”).
- ADA detection differences (Fig 3A, p. 5) are not statistically significant after 24 months (p=0.145), but the narrative text makes it sound more impactful than the stats justify. Tone this down.
- Discussion
It’s a bit long and could be more focused. Consider separating into clear subsections:
- Main finding i.e., low concentrations <1 mg/L linked to flare risk.
- Pharmacokinetics i.e., half-life differences explain why ADL shows stronger association.
- Immunogenicity i.e., MTX tapering increases ADA detection but not concentrations.
- Strengths & limitations i.e., avoid burying these points in the middle.
Limitations section is solid but could highlight patient-reported outcomes (burden worsens before DAS-based flare) more strongly.
- Figures & Tables
- Figures 1&2 : Nicely done but the Y-axis is logarithmic…………this should be explicitly stated in captions for clarity.
- Figure 3: Good representation of ADA trends, but legends could be simplified. Currently a bit technical for readers outside rheumatology.
- Table 2 (p. 6): Excellent summary of odds ratios. But visually, bolding significant p-values is good; maybe add shading or italics for clarity.
- Conclusions
- Avoid over-generalizing: “Our findings could serve as a basis for future research” is safe, but avoid suggesting exact thresholds for clinical practice yet, given variability and limitations.
Some recent and suitable references can be added to support these are suggested below.
- https://doi.org/10.47278/journal.abr/2024.003
- https://doi.org/10.47278/journal.ijab/2024.093
very minor
Reviewer 5 Report
Comments and Suggestions for Authors
The manuscript presents a valuable post-hoc analysis of the TARA trial, investigating the critical relationship between serum drug concentrations of adalimumab (ADL) and etanercept (ETN) and the occurrence of disease flares in RA patients undergoing treatment tapering. The study's strengths include its design, the use of a drug-tolerant ADA assay, and its focus on a clinically relevant question. The findings suggest a potential target concentration for maintaining remission during tapering and provide novel insights into immunogenicity in long-term treatment.
- The manuscript states that 111 patients were selected based on strict protocol adherence and having >3 samples. A flowchart (Supplementary Figure S1) is referenced but not provided for review. Including this flowchart in the main text or as an accessible supplement is crucial to assess selection bias and understand the representativeness of the analyzed cohort relative to the original TARA trial population 7.
- The flare definition (DAS>2.4 or SJC>1) is noted to be based on expert opinion. While valid, briefly citing a reference or guideline that supports this definition would enhance its credibility. Alternatively, a sensitivity analysis using an alternative definition could be mentioned to demonstrate robustness.
- The manuscript correctly identifies non-trough sampling as a limitation, as it introduces significant variability in drug level measurements. To mitigate this concern, consider adding a sentence in the methods or discussion stating whether the time of blood draw relative to the last injection was recorded and if any statistical adjustments were attempted to account for this variability.
- The observation that increased ADA detection did not correlate with lower ADL levels is fascinating and is aptly explained by the use of a drug-tolerant assay. This is a key insight. Elaborate slightly in the discussion on the clinical implications: this suggests that in patients on long-term therapy, not all detected ADAs are neutralizing or clinically significant, reinforcing the guideline to only test drug levels and ADAs on indication 210.
- The conclusion that flares occur below 1 mg/L is supported by the data (median 0.44 mg/L for ADL, 0.05 mg/L for ETN). However, the IQRs are wide (up to 1.15-1.29 mg/L). The text should more precisely state that the medianconcentration at flare was below 1 mg/L, but a range of concentrations were observed, suggesting patient-specific factors are also at play.
- The GEE analysis found no significant association between ETN levels and flare risk, which contrasts with ADL. The discussion attributes this to ETN's shorter half-life and the 3-month sampling interval, which is a reasonable explanation. Strengthen this argument by explicitly stating that the rapid decline of ETN means many patients without flares also quickly have low levels, making it statistically difficult to find a differential association.
- The finding that tapering MTX first increased ADA detection without affecting serum concentrations is clinically important. Discuss this in the context of de-escalation strategies. It suggests that in patients in sustained remission on combination therapy, tapering MTX first may be a viable strategy without immediately compromising ADL efficacy, potentially reducing MTX-related side effects.
- The manuscript mentions that most ADAs against ADL are neutralizing antibodies (NAb) 10. Since the drug-tolerant assay detects all ADAs, briefly speculate whether the ADAs detected after MTX tapering in this long-term cohort might be predominantly non-neutralizing or low-affinity, explaining the lack of impact on drug levels.
- The explanation of the half-life and clearance differences between ADL (a monoclonal antibody) and ETN (a receptor fusion protein) is good. To enhance it, consider adding a sentence on the role of the neonatal Fc receptor (FcRn) in recycling ADL and extending its half-life, a mechanism that differs for ETN 5.
- Ensure units for drug concentrations are consistent. The Abstract uses mg/L, which is correct. Double-check that all instances in the main text, figures, and tables also use mg/L and not μg/mL (1 mg/L = 1 μg/mL, but consistency is key for readability).
- For the ADA analysis (Figure 3A), the manuscript states the difference was not significant at 24 months (p=0.145). Clarify what statistical test was used (e.g., Chi-square or Fisher's exact test) for this comparison in the figure legend or methods section.
- The GEE analysis categorized drug levels into tertiles ("low," "medium," "high"). Briefly justify in the methods why categorization was chosen over using concentration as a continuous variable (e.g., due to a non-linear relationship observed in preliminary analyses).
- The study population consists of well-controlled RA patients on combination therapy for a median of ~6 years. Explicitly discuss the generalizability of the findings to other populations, such as patients with shorter disease duration, those on monotherapy, or patients with other diseases treated with TNF inhibitors.
- The introduction effectively sets the stage but could more sharply highlight the knowledge gap this study fills. Add a sentence specifically stating that while TDM is recommended, the minimum effective concentration during tapering—a key parameter for successful de-escalation—is poorly defined and that this study aims to address that.
- The introduction outlines the aims well. To make it even stronger, consider adding a sentence at the end previewing the main finding: e.g., "We hypothesized that the majority of flares would occur as serum drug concentrations fall below a specific threshold, and that immunogenicity would be influenced by the order of tapering."
- The objectives in the abstract are 1) assess association between concentration and flare, and 2) explore impact of MTX tapering on immunogenicity. The results are well-aligned. Ensure the discussion sequentially addresses these points, first discussing the flare concentration and then the immunogenicity findings.
- The discussion is comprehensive. To make the take-home messages even clearer for clinicians, consider adding a short subsection titled "Clinical Implications" or "Conclusions and Future Directions" that succinctly summarizes the key practical findings and proposed next steps.
- There are minor typos/formatting (e.g., "vice and versa" in Introduction, "C-reactief protein" in Table 1). A thorough proofread is recommended to catch these small errors and ensure professional presentation.
- The conclusion mentions this could be a basis for future research. Be more specific. Suggest future studies could prospectively test a TDM-guided tapering strategy versus standard care, or investigate the characteristics of ADAs detected by drug-tolerant assays in long-term treatment.
Round 2
Reviewer 2 Report
Comments and Suggestions for Authors
- There are few grammatical mistakes, which need to be addressed. It is better to check the whole manuscript only for language corrections and typos.
- In the introduction, the authors stated about RA with recent article with DMARDs, Biologic DMARDs, Pathophysiology and nanotechnology emerging discipline, so provide some examples of RA based nanotechnology delivery or nanomedicine with their limitations by highlighting the authors' work.
- The rationale for developing this specific delivery system for RA could be stated more clearly upfront. Why was this particular approach chosen over other RA therapy.
- A more thorough discussion of potential limitations of the approach and study design would provide a more balanced perspective. The paper could benefit from a clearer discussion of next steps and future research directions to build on this work.
- Provide some explanation of adalimumab(ADL) and etanercept(ETN) and MTX in RA with citations, which means why the authors have chosen adalimumab (ADL) and etanercept(ETN) and MTX.
- While the study compares the adalimumab(ADL) and etanercept(ETN) and MTX, it doesn't compare it to other current treatments for rheumatoid arthritis. Including such comparisons would provide better context for the potential clinical significance of this treatments for rheumatoid arthritis.
-
The study doesn't explore different dosing regimens. Optimizing the dose could potentially improve efficacy or reduce side effects.
- An economic analysis comparing the potential cost of this Biologic DEMARDs and Conventional DEMARDs to current treatments would be useful for assessing its practical viability.
- The study doesn't mention whether both male and female patients were used, or if any gender differences in response were observed.
- The authors clearly define that how the therapeutic drug monitoring improve the clinical decision-making during biologic drug tapering in RA and also the authors please mentioned methodological challenges in studying serum adalimumab (ADL) and etanercept(ETN) and MTX and flare prediction in biologic DEMARDs and DEMARDs tapering in RA for future researcher.
Reviewer 4 Report
Comments and Suggestions for Authors
acceptable for publication
Reviewer 5 Report
Comments and Suggestions for Authors
This well-conducted study provides valuable and clinically relevant insights into the pharmacokinetics of TNF-inhibitors during tapering. The findings clearly demonstrate that serum concentrations of both adalimumab and etanercept fall below 1 mg/L at the time of a flare, offering a practical threshold for clinicians. The observation that methotrexate tapering increases immunogenicity without affecting drug levels is a novel and important contribution to the field. The manuscript is strengthened by its robust design and clear results, which successfully answer the research objectives. It is recommended for acceptance.
Round 3
Reviewer 2 Report
Comments and Suggestions for Authors
The authors did a good job in revising the manuscript titled "The association between serum drug concentration and a flare in rheumatoid arthritis patients tapering TNF inhibitor". The revised version has been extensively improved. I can recommend this manuscript for publication in Pharmaceuticals journal.
